# Effects of Surface Chemical Modification by Ethoxysilanes on the Evolution of 3D Structure and Composition of Porous Monoliths Consisting of Alumina Hydroxide Nanofibrils in the Temperature Range 25–1700 °C

**DOI:** 10.3390/nano12203591

**Published:** 2022-10-13

**Authors:** Anatole Khodan, Andrei Kanaev, Mikhail Esaulkov, Mikhail Kiselev, Victor Nadtochenko

**Affiliations:** 1A.N. Frumkin Institute of Physical Chemistry and Electrochemistry Russian Academy of Sciences, 199071 Moscow, Russia; 2N.N. Semenov Federal Research Center for Chemical Physics, Russian Academy of Sciences, 119991 Moscow, Russia; 3Laboratoire des Sciences des Procédés et des Matériaux, CNRS, Université Sorbonne Paris Nord, 93430 Villetaneuse, France; 4Femtonica LLC, 108840 Moscow, Russia

**Keywords:** mesoporous nanomaterials, aluminum oxyhydroxides, 3D nanostructure, surface chemical modification, diffusion, phase transitions

## Abstract

Bulk nanomaterials with an open porosity offer exciting prospects for creating new functional materials for various applications in photonics, IR-THz optics, metamaterials, heterogeneous photocatalysis, monitoring and cleaning toxic impurities in the environment. However, their availability is limited by the complexity of controlling the process of synthesis of bulk 3D nanostructures with desired physicochemical and functional properties. In this paper, we performed a detailed analysis of influence of a silica monolayer chemically deposited on the surface of a monolithic ultraporous nanostructure, consisting of a 3D nanofibril network of aluminum oxyhydroxide, on the evolution of structure and morphology, chemical composition and phase transformations after heat treatment in the temperature range of 20−1700 °C. The experimental results are interpreted in the framework of a physical model taking into account surface and volume mass transport and sintering kinetics of nanofibrils, which made it possible to estimate activation energies of the surface diffusion and sintering processes. It is shown that the presence of a surface silica monolayer on the surface affects the kinetics of aluminum oxyhydroxide dehydration and inhibits diffusion mass transfer and structural phase transformations. As a result, the overall evolution of the 3D nanostructure significantly differs from that of nanomaterials without surface chemical modification.

## 1. Introduction

The development of methods for the synthesis of 3D nanostructures with desired physical and chemical properties is an important step in the creation of new functional nanomaterials and nanocomposites for a variety of applications: heterogeneous photocatalysis, biodegradation of microorganisms and purification of air and water media, elements of optics and waveguides for the IR–GHz range, antenna arrays, metamaterials, etc. 

The formation of fibrous aluminum oxyhydroxides at the surface of a liquid metal solution of aluminum in mercury Hg(Al) was described first by W. Wislicenus [1,2], and the composition and structure of these fibrous forms of alumina have been described by M. R. Pinnel and J. E. Bennett [3]. J. L. Vignes et al. [4] have succeeded in improving the preparation process and optimizing the conditions for controlled growth of porous monolithic 3D nanomaterials on a thin Hg layer deposited on the surface of a metallic Al plate [4,5]. The use of their laboratory technology made it possible to grow monolithic blocks of a highly porous 3D nanomaterial based on aluminum oxyhydroxides (PMOA) up to 1 L in volume, with a constant cross-section shape and controlled height [5,6].

At present, the growth of 3D nanostructures of aluminum oxyhydroxides, similar to PMOA, has been experimentally confirmed for the surface oxidation in humid inert gas and air for liquid metal alloys Me(Al), where Me = Hg, Ga, In, Sn, Bi and Pb [7,8]. Moreover, the growth of 3D blocks of ordered co-aligned alumina nanofibrils during surface oxidation of aluminum melt in a humid argon atmosphere was also observed and confirmed [9,10].

However, the mechanism of the formation of monolithic highly porous 3D nanostructures as well as the 1D growth of aluminum oxyhydroxide nanofibrils during the surface oxidation of liquid Me(Al) alloys in a humid gas or air remain insufficiently studied to date. It has been established that the basic element of the 3D structure of PMOA nanomaterials is the nanofibrils of aluminum oxyhydroxide, which have a diameter *d*_0_ from 5 to 9 nm and average length *a*_0_ in the range of 100 to 300 nm, resulting in the average ratio of fibril length to diameter (shape factor) of *a*_0_/*d*_0_ ≈ 30. The density of monolithic raw PMOA samples grown at 25 °C is *ρ*_0_ = 0.018 ± 0.005 g∙cm^−3^ [11]. The annealing of PMOA materials in the temperature range up to 1700 °C does not cause any destruction or cracking of the material, but the linear dimensions of the samples decrease isotropically, and the bulk density increases more than 160 times up to 2.9 ± 0.1 g∙cm^−3^ (4 h at *T* = 1600 °C). This very important technological property makes it possible to change not only the density of PMOA materials but also the value of open porosity in the range from 99% to ~25%. A fairly complete description of the properties of PMOA materials is given in References [11,12], where a quantitative physical model describing the evolution of the 3D structure of PMOA materials upon annealing in the temperature range up to 1700 °C also has been proposed. 

The physical and chemical properties of raw PMOA materials are largely determined by the features of their 3D structure, possessing completely open mesoporosity and a large specific surface area of ~300 m^2^/g, which can be increased up to ~750 m^2^/g using cryogenic drying.

The practical use of PMOA-based materials largely depends on their mechanical strength. Using annealing can control the properties of PMOA materials and significantly increase their strength due to the overall compaction, which is due to a significant change in morphology: the shape of nanofibrils approaches a spheroid, and the shape factor decreases by ~10 times. However, these changes affect a very important parameter for creating functional heterostructures, the specific surface area. Annealing of PMOA during 4 h at 1500 °C decreases its specific surface area to ~1 m^2^/g. It has been shown [11,12] that diffusion transport controls kinetics of the nanofibril morphology modification and parameters of 3D nanostructure, with the surface diffusion playing the main role during annealing of PMOA materials up to ~900 °C and the bulk diffusion dominating at higher temperatures. 

The aim of this work was to reveal and study the effect of the SiO_2_ monolayer deposited on the nanofibril surface by the hydrolysis of ethoxysilanes, which is expected to affect the diffusion mass transport and kinetics of morphological and structural phase transformations in PMOA-based materials after annealing in the temperature range 25–1700 °C. This work continues our studies of the synthesis methods of monolithic porous 3D nanomaterials based on aluminum oxyhydroxides and evolution of physical and chemical properties during annealing at temperatures up to 1700 °C [11,12].

## 2. Materials and Methods

*Preparation and properties of raw PMOA samples*. Monolithic samples of raw PMOA with a volume of *V*_0_ ≤ 8 cm^3^ were grown at a thin layer (~1 μ) of a liquid Hg(Al) alloy deposited on the surface of an Al (99.99% purity) metal plate. PMOA samples were grown in a climatic WIESS VK3-180/40 chamber in an air atmosphere with controlled humidity ~75% and temperature ~25 °C; the growing conditions were similar to those reported earlier [4,6,11].

The oxidation products have the shape of fibers, 1D nanofibrils, growing to the surface. During growth the side contacts between nanofibrils are formed, and a highly porous monolithic 3D structure of PMOA grows with the rate of ~1 cm/h, consisting of entangled fibrils with an average diameter *a*_0_ ≈ 4–10 nm and length *d*_0_ ≈ 150–350 nm [6,11,12]. In general, the oxidation reaction of Hg(Al) alloy surface in humid air can be presented as:2Hg(Al)_surf._ + (*n* + 1)∙H_2_O + O_2_ → Al_2_O_3_∙*n*H_2_O + 2(Hg)_surf._ + H_2_↑,(1)
where the product of oxidation Al_2_O_3_∙*n*H_2_O is amorphous hydrated alumina, for which the *n* value can vary from 3.4 to 4.6 depending on the ambient humidity. For raw PMOA grown under standard conditions, the average experimental value: *n* ≈ 3.6 ± 0.2. Table 1 summarizes main physical properties of raw PMOA samples with the composition Al_2_O_3_∙3.6H_2_O.

In previous studies, the main features of the chemical state of raw PMOA materials have been established [6,11,12]. NMR methods showed that Al^+3^ cations in raw PMOA samples have different anionic environments:83% for octahedral, 16% for pentahedral, and 1% for tetrahedral. [13]. It is known that Al^+3^ cations in the octahedral environment are able to unite and form polymolecular chains, as shown in Figure 1a [14]. The elements of molecular structure and water molecules are linked by hydrogen bonds to form an amorphous polymolecular multinuclear structure of PMOA. The observed growth texture of PMOA [6,7,8,9] is probably due to the mechanism of 1D growth of ABx molecular chains [14] associated with the formation of nanofibrils, the main element of the 3D network of the nanostructure, as shown in Figure 1b. Based on the obtained results, it can be concluded that the PMOA raw materials are monolithic mesoporous nanostructures formed as a three-dimensional network of nanofibrils in the amorphous state with elements of the pre-boehmite structure with the composition Al_2_O_3_∙*n*H_2_O, where the value of *n* can vary from 3.2 to 4.6 depending on the synthesis conditions and air humidity.

*Surface chemical modification of raw PMOA samples.* After growing, raw PMOA samples were placed in a sealed glass container and kept in saturating vapor of trimethylethoxysilane (CH_3_)3SiOCH_2_CH_3_ (TMES) or methyltrimethoxysilane CH_3_Si(OCH_3_)_3_ (MTM) at room temperature ~22–24 °C. The molecules of ethoxysilanes hydrolyzed due to an interaction with the hydrated alumina, which results in the formation of chemically adsorbed groups at the PMOA surface: [=Al–O–]Si–(CH_3_)|_surf._ in the case of TEMS or [=Al–O–]_3_Si–(CH_3_)|_surf._ for MTM.

Kinetics of surface modification at ~22 °C in saturated MTM vapor can be estimated using the weight gain data: After 4 h of treatment, about 79% of the PMOA surface is modified to the state [=Al–O–]_3_Si–(CH_3_)|surf.; this corresponds to an increase in the silicon content up to 9.96 wt.%. Furthermore, it was found that under normal conditions the maximum value ~ 93% of surface saturation in the MTM vapor can be achieved with increasing processing time *t* ≥ 7 h. The saturation criterion was the cessation of weight gain within the measurement error.

Similar results were obtained for the treatment in TMES vapor. Previously it has been established that PMOA samples placed in TMES vapor for 1 h accumulate ≈7wt.% of silicon [5,15]. The weight gain of the samples became small after ≥3.5 h at 25 °C, and the fibril surfaces became saturated with the hydrolysis products. It should be noted that the process of the PMOA surface saturation with the TMES hydrolysis products proceeds almost 2 times faster than in MTM vapor.

It is important to note that the PMOA-M samples chemically modified in TMES or MTM vapor after annealing at *T* ≥ 800 °C have approximately the same number of SiO_2_ molecules, constituting ~3% of the samples mass. By taking into account the specific surface, the deposition of almost one monolayer of SiO_2_ at the surface can be estimated.

In this work, isochronous 4 h annealing at a constant temperature was used for all PMOA and PMOA-M samples, unless another heat treatment time was specified.

*Instruments and methods of research*. A comparative study of chemical composition and physical properties of PMOA and PMOA-M samples was performed using thermal analysis (TA) in an argon or nitrogen environment using DSC-Q100 and TGA-Q500 (TA Instruments, New Castle DE, USA) in dynamic and modulation modes and a TGA-Q50 equipped with a Nicolet iS 10 FTIR Spectrometer (Thermo Fisher Scientific, Waltham MA, USA).

A sample with mass of about 20 mg was heated in argon flow (~50 mL/min) at the increment rates between 0.1 and 50 °C/min up to the maximum temperature of 1000 °C, as shown in Figure 2. TGA data also confirmed that at room temperature the complete surface saturation of PMOA monolith with the hydrolysis products of ethoxysilanes was achieved after ≤4 h.

The morphology and 3D structure features of the samples were characterized using a JEOL JSM 6060 scanning electron microscope. The obtained SEM images clearly show significant differences in both the morphology of nanofibrils and the overall 3D nanostructure between the PMOA and PMOA-M samples after treatment at T > 500 °C and higher.

The structure and phase composition of PMOA and PMOA-M samples were studied using powder diffractometers INEL XRG 3000 and Bruker D8 with a Cu-Kα (*λ* = 0.15418 nm) X-ray source. Neutron scattering studies were carried out using small-angle KWS-2 and ultra-small-angle KWS-3 diffractometers in Research Neutron Source Heinz Maier-Leibnitz, reactor FRM II [12].

## 3. Results and Discussion

### 3.1. Chemical Composition Transformations in PMOA and PMOA-M Samples

The presence of silicon compounds on the PMOA-M surface has a significant effect on the change in physicochemical properties relative to the initial raw PMOA. It should be noted that PMOA-M samples after chemical modification acquire hydrophobic properties [5], but the most significant differences in the evolution of the morphology and 3D nanostructure of the studied samples were observed after thermal treatment.

It should be noted that we always used isochronous 4 h annealing, unless other annealing conditions are specified.

Annealing of PMOA-M samples at *T* ≥ 450 °C led to a complete desorption of methyl groups, which can be used to estimate the number of SiO_2_ molecules bound on the surface and to express it as a monolayer (ML) fraction: *h* = 0.96 ± 0.05. The presence of a monomolecular layer of SiO_2_ at the surface of 3D structure has an important influence on the kinetics of the chemical composition changes and on the onset of structural-phase transformations of PMOA-M samples. For example: the maximum loss of structural water during the annealing (below onset of γ-Al_2_O_3_ crystallization) is achieved at 450 ± 20 °C for PMOA samples, whereas such dehydration degree of PMOA-M samples is achieved at temperatures ≥840 °C. The results of thermal analysis of raw PMOA and PMOA-M samples and those preliminarily treated at a fixed temperature in the range between 100 to 1200 °C with subsequent cooling to room temperature are shown at Figure 2.

The treatment in ethoxysilane vapor led to partial dehydration of raw PMOA, and a surface layer consisting of –Si(CH_3_)x was formed. The surface reaction for TMES can be represented as:=Al–OH|_surf._ + C_2_H_5_O–Si≡(CH_3_)_3_ → =Al–O–Si≡(CH_3_)_3_|_surf._ + C_2_H_5_OH↑,(2)
or in the case of reaction with MTM vapor:3(=Al–OH|_surf._) + (CH_3_O)_3_≡Si−CH_3_ → (=Al–O)_3_≡Si–(CH_3_)|_surf._ + 3(CH_3_OH)↑.(3)

After surface chemical modification and annealing at temperatures ≤ 450 °C, the chemical composition of PMOA-M can be represented as: Al_2_O_3_∙*n*H_2_O∙*x*Si∙*y*(CH_3_). The annealing in air at higher temperatures of 450–950 °C led to decomposition/oxidation of hydrolysis products, and a thin layer of SiO_2_ remained at the surface of aluminum oxyhydroxide. Under these conditions, the composition of PMOA-M samples can be represented as: Al_2_O_3_∙*n*H_2_O∙*x*(SiO_2_), where *n* → 0. Short-term annealing for 1 h at *T* ≥ 450 °C was quite sufficient for removal of methyl groups from the PMOA-M surface, and the SiO_2_ layer formed at the nanofibril surface with the thickness estimated as 0.96 ± 0.05 ML.

An almost continuous layer of SiO_2_ formed on the surface prevented the free dehydration of aluminum oxyhydroxides during heating. This can be clearly seen from a comparison of TGA results for untreated PMOA [11] and chemically modified PMOA-M materials in the low-temperature range ≤200 °C (Figure 2).

The TGA results allowed the water content to be estimated in PMOA-M samples of the general chemical composition of Al_2_O_3_∙*n*H_2_O∙*x*(SiO_2_) and revealed changes in the contents of “structural water” *n*_str_ and “adsorbed water” *n*_ads_ (*n* = *n*_str_ + *n*_ads_) in samples subjected to the preliminary annealing. The values of *n*, *n*_ads_ and *n*_str_ obtained at different pre-annealing temperatures are given in Table 2. We notice that *n*_ads_ does not change significantly, and a small difference between these data fits the confidence range of analysis.

It should be noticed that changes in the chemical composition and structure of raw PMOA samples began almost immediately upon heating to *T* ~ 100 °C, while for PMOA-M samples, similar changes in the chemical composition and phase state required heating up to much higher temperatures of 380–650 °C (Figure 2).

The TEMS treatment has a strong effect on the structure and kinetics of phase transformations in PMOA-M samples. The presence of a thin layer of amorphous SiO_2_ on the surface of PMOA-M delays the loss of structural water from aluminum oxyhydroxide, limits diffusion mobility on the surface of nanofibrils, and inhibits the structural phase transition from amorphous alumina into *γ*-Al_2_O_3_, what becomes especially noticeable at temperatures ≥ 950 °C. These results are discussed in Section 3.3.

The evolution of the overall 3D nanostructure of PMOA-M materials and changes in nanofibril morphology have been studied by SEM methods and are discussed in detail in Section 3.3.

### 3.2. The Structure and Phase Transformations in PMOA and PMOA-M Samples

The structural changes and phase transformations in PMOA and PMOA-M samples were studied using X-ray diffraction methods in the temperature range of 100 to 1600 °C after isochronous 4 h annealing (Figure 3).

PMOA-M samples in the amorphous state of Al_2_O_3_∙*n*H_2_O∙*x*(SiO_2_) manifested the transition to the γ-Al_2_O_3_ phase at T ≥ 870 °C, which matches the onset transition temperature in PMOA (Al_2_O_3_∙nH_2_O). However, a significant difference in the behavior of PMOA-M and PMOA samples appeared at elevated temperatures ≥ 1150 °C. In PMOA-M the *θ*-Al_2_O_3_ phase formation onset is seen, and this structural state remains stable up to ~ 1400 °C, after which crystallization of the *α*-Al_2_O_3_ phase can be observed. In contrast, in PMOA samples, the crystallization of the *α*-Al_2_O_3_ phase begins much earlier at a temperature of ~ 1150 ° (see Figure 3 and Table 3). It is important to notice that after sintering at 1400 °C, the phase state of *α*-Al_2_O_3_ in PMOA-M differs from PMOA: additional patterns appear that can be attributed to mullite: ~(1.5 ÷ 2)Al_2_O_3_·SiO_2_, as shown in Figure 4.

### 3.3. Evolution of 3D Nanostructure and Volume Density in PMOA and PMOA-M Samples

A strong influence of the surface chemical modification on morphology and 3D nanostructure of PMOA-M samples after annealing at 1400 °C is shown in Figure 5. The same effects of the strong influence of a thin SiO_2_ surface layer were noted earlier [5,15].

The isochronous annealing at temperatures from 100 to 1600 °C did not cause a destruction of monolithic samples: the general 3D structure of PMOA-M samples was preserved, while the volume density increased as reported previously in pure PMOA [11]. This is clearly seen when comparing the experimental data presented in Figure 6.

Taking into account the results obtained for PMOA and PMOA-M samples and comparing them in terms of the kinetics of the chemical composition changes (Figure 2), the volume density evolution (Figure 6) and structural and phase transformations during annealing (Figure 3) allow us to state that even single monolayers of SiO_2_ on the surface of PMOA-M lead to: (1) a noticeable inhibition of the dehydration of nanofibrils during annealing; (2) the onset of phase transformations in PMOA-M shifts towards higher temperatures, in particular, the difference exceeds 250 °C for the θ-Al_2_O_3_ → α-Al_2_O_3_ phase transition during isochronous 4 h annealing (Table 3); (3) in the temperature range up to 1200 °C, the surface diffusion in PMOA-M samples is much lower than that of an open surface in PMOA. All of these affect the overall kinetics of structural changes, minimizing the morphology changes of nanofibrils, and the 3D nanostructure remains close to the initial state (Figure 5).

One of the main objectives of this study was a quantitative assessment of the physical parameters that determine the mechanism and kinetics of evolution of the 3D nanostructure PMOA-M in the temperature range 25 ÷ 1650 °C. For this purpose, we used the physical model proposed earlier to describe the effect of annealing on the three-dimensional structure of PMOA [11]. The following physical parameters for PMOA-M samples were used:

*m*_raw_—total mass of the sample;

*V*_raw_—the sample volume;

*ρ* = *m*_raw_/*V*_raw_—the density of bulk PMOA-M.

To determine the volume density of PMOA-M materials, the parameters of the elements that form a three-dimensional nanostructure can be used:(4)$$\rho_{0}=\frac{m_{raw}}{V_{raw}}=\frac{\rho_{f,\;0}}{V_{raw}}\left(\frac{\pi d_{0}{}^{2}}{4}\right)l_{total}=n_{f}\rho_{f,\;0}\frac{\pi }{4}\frac{d_{0}^{2}a_{0}}{a_{0}^{3}}=n_{f}\rho_{f,\;0}\frac{\pi }{4}{\left(\frac{d_{0}}{a_{0}}\right)}^{2}$$
where $\rho_{f,0}$, the density of fibril material, is around the skeletal density of raw PMOA: 2.25 ± 0.05 g∙cm^−3^, as shown in Table 1, which is slightly less than the density of Al(OH)_3_ monoclinic crystals: 2.42 g∙cm^−3^ for hydrargillite and 2.53 g∙cm^−3^ for bayerite.

$a_{0}$—the average fibril length: $a_{0}=\frac{1}{n}\overset{n}{\underset{i=1}{\sum }}a_{i}$.

$d_{0}$—the average fibril diameter: $d_{0}=\frac{1}{n}\overset{n}{\underset{i=1}{\sum }}d_{i}$;

$l_{total}$—the total length of all fibrils located in the volume $V_{NOA}$;

$n_{f,\;0}=\frac{l_{total}\;}{a_{0}}$—the average number of fibrils with a diameter $d_{0}$ and an average length $a_{0}$, which are located in the volume $a_{0}^{3}$.

The density $\rho \left(T,t\right)$ dependence on temperature and annealing time in the high temperature range ≥1200 °C, where the sintering and agglomeration processes are started, and volume density evolution at Figure 6b can be described by the expression:(5)$$\rho \left(T,t\right)=\rho_{f}\frac{\pi ad^{2}}{4a^{3}}=\rho_{f}\frac{\pi }{4}{\left(\frac{d_{0}}{a_{0}}\cdot \frac{1}{\sqrt{{\left(1-\frac{\sqrt{Dt}}{a_{0}}\right)}^{3}}}\right)}^{2}=\rho_{f}\frac{\pi }{4}\frac{{\left(\frac{d_{0}}{a_{0}}\right)}^{2}}{{\left(1-\frac{\sqrt{Dt}}{a_{0}}\right)}^{3}},$$
where $D=D_{0}\exp\left[-\frac{E_{D}}{k_{B}T}\right]$ is the ”effective” surface diffusion coefficient, and $E_{D}$ is the activation energy of surface diffusion.

Isochronous annealing in the temperature range *T* ≥ 1200 °C activates bulk diffusion in PMOA-M, and the process of sintering of the porous 3D structure begins, which can be described by the simplified version of the Ivensen equation [16]. We assume that the changes of free volume in time follow the relationship:(6)$$\frac{dV}{dt}=-BV,\;\mathrm{were}\;B=B_{0}\cdot \exp\left(-\frac{E_{b}}{RT}\right)$$

Using (5), we can obtain the dependence of mass density $\rho \left(T,t\right)$ on temperature and annealing time:(7)$$\rho \left(T,t\right)=\frac{\rho_{f}}{1+\frac{\rho_{f}}{\rho_{0}}exp\left(-B_{0}t\cdot \exp\left(-\frac{E_{b}}{RT}\right)\right)}$$

It is important to note that in the high temperature range ≥1000 °C, the chemical composition approaches ~Al_2_O_3_∙H_2_O∙*x*(SiO_2_), as shown in Figure 2, and the skeletal density of PMOA-M in this case is close to the corresponding orthorhombic AlOOH phases: 3.01 g∙cm^−3^ for boehmite and 3.44 g∙cm^−3^ for diaspore.

Using the experimental data (Figure 6a,b), the main parameters of 3D structure of PMOA and PMOA-M materials were obtained for the temperature range of 25 to 1700 °C, as shown in Table 4.

### 3.4. Evolution of the Specific Surface Area and Thickness of Surface SiO_2_ Layer in PMOA and PMOA-M Samples

Raw PMOA samples modified with TMES after annealing at *T* ≥ 1000 °C possessed a specific surface area of ~204 m^2^/g, which is noticeably larger than that of the unmodified samples: ~170 m^2^/g. This difference increases at the annealing temperature up to 1300 °C, and the specific area ratio PMOA-M/PMOA increases by ~11 times. In contrast, the difference in the specific surface area begins to decrease at the annealing temperatures ≥ 1600 °C, which can be easily explained by an effective sintering process of both PMOA and PMOA-M materials (Figure 7).

The amount of amorphous surface SiO_2_ on PMOA-M changed during annealing from the initial value, ~1 monolayer (ML) at *T* < 800 °C, to ~500 ML after annealing at 1600 °C. Despite several phase transformations, including amorphous Al_2_O_3_∙3.6H_2_O, γ-, θ- and α-Al_2_O_3_ (depicted in Figure 3) and significant changes in the morphology of the 3D structure, the SiO_2_ layer was preserved on the surface of nanofibrils, which after prolonged annealing (Figure 8) could include silicoaluminates [15,17]. In order to describe the changes in thickness of SiO_2_ layer upon annealing in the temperature range of 25 to 1600 °C, the earlier proposed 3D model was applied [11].

The number of SiO_2_ molecules deposited per unit surface PMOA may be estimated as:(8)$$n_{S}^{SiO_{2}}=\frac{m_{total\;}^{ads}.k_{SiO_{2}}^{ads}}{M_{SiO_{2}}}.\frac{N_{A}}{S_{total}^{ads}}=\frac{N_{A}}{M_{SiO_{2}}}.\frac{k_{SiO_{2}}^{ads}}{S_{total}^{NOA}}\left[\frac{m_{total}^{PMOA-M}}{m_{total}^{PMOA}}-1\right]$$
where$\;m_{total}^{ads}$ = $m_{total}^{PMOA-Si}-m_{total}^{PMOA}$ is the mass difference between the raw PMOA and PMOA-M samples;

$m_{SiO_{2}}$—molecular weight of SiO_2_;

$S_{total}^{ads}$—the specific surface area of adsorbate (PMOA);

$m_{total}^{ads}$—molecular weight of the adsorbed products of MTM hydrolysis: ≡Si–(CH_3_)|_surf._.

Similarly, we can define $m_{S}^{SiO_{2}}$ as the mass SiO_2_ molecules per unit area:(9)$$m_{S}^{SiO_{2}}=\frac{k_{SiO_{2}}^{ads}}{S_{total}^{PMOA}}.\left(\frac{m_{total}^{PMOA-M}}{m_{total}^{PMOA}}-1\right)$$
and $h_{S}^{SiO_{2}}$: the thickness of the SiO_2_ layer on the surface of the PMOA-M:(10)$$h_{S}^{SiO_{2}}=\frac{k_{SiO_{2}}^{ads}}{\rho_{SiO_{2}}S_{S}^{PMOA}}.\left(\frac{m_{total}^{PMOA-M}}{m_{total}^{PMOA}}-1\right)$$

The resulting value $h_{S}^{SiO_{2\;}}$can be attributed to the size of molecules SiO_2_ and estimate the number of monolayers (ML) on the surface of the material, depending on the changes in the specific surface during annealing. For quantitative estimates, we used the following values: the volume of SiO_2_ molecule: *V* = 0.038 nm^3^; the surface area covered by SiO_2_ molecules: *S* = 0.1364 nm^2^; diameter of SiO_2_ molecules $d_{SiO_{2}}$: minimum 0.302 nm, maximum 0.372 nm. The estimate of the density in the SiO_2_ monolayer can be derived from the relationship:(11)$$ML_{SiO_{2}}≅\rho_{SiO_{2}}.d_{SiO_{2}}$$

The number of SiO_2_ monolayers on the PMOA-M surface can be estimated:(12)$$n_{ML}^{SiO_{2}}=\frac{k_{SiO_{2}}^{ads}}{S_{toal}^{PMOA}}.\left(\frac{m_{total}^{PMOA-M}}{m_{total}^{PMOA}}-1\right).\left[\frac{1}{\rho_{SiO_{2}}.d_{SiO_{2}}}\right]=\frac{k_{SiO_{2}}^{ads}}{S_{total}^{PMOA}.\rho_{SiO_{2}}.d_{SiO_{2}}}\left(\frac{m_{total}^{PMOA-M}}{m_{total}^{PMOA}}-1\right)$$

Or:(13)$$n_{ML}^{SiO_{2}}=\frac{h_{S}^{SiO_{2}}}{d_{SiO_{2}}}=\frac{k_{SiO_{2}}^{ads}}{S_{total}^{PMOA}.\rho_{SiO_{2}}.d_{SiO_{2}}}\;\left(\frac{m_{total}^{PMOA-M}}{m_{total}^{PMOA}}-1\right)$$

To estimate the number of monolayers $n_{ML}^{SiO_{2}}$ on the PMOA-M surface depending on the treatment time in TMES vapors and annealing temperature, we used expression (13) and the specific surface area results of the PMOA and PMOA-M samples (Figure 7 and Figure 9a). The dependences obtained for $n_{ML}^{SiO_{2}}$ describe well the experimental data presented in Figure 9b.

## 4. Conclusions

Effects of the surface deposition of a thin SiO_2_ surface layer on structural properties and modification kinetics of ultraporous 3D aluminum oxyhydroxide materials were investigated in the temperature range from ~25 up to 1700 °C. The isochronous 4 h annealing did not lead to any damage of the material microstructure, which open porosity was preserved. At the same time, the volume density increased by a factor of ~10^2^, specific surface area decreased from ≥300 to ~2–4 m^2^/g and porosity decreased from ~99.5 to 25%. We showed that the surface SiO_2_ layer affects the chemical composition [Al_2_O_3_∙*n*H_2_O]∙*x*(SiO_2_) of the material and, in particular, contents of structural and adsorbed water and leads to the inhibition of the phase transition process: from amorphous [Al_2_O_3_∙*n*H_2_O] to crystalline γ-, θ- and α-Al_2_O_3_ materials. In particular, onset of the α-Al_2_O_3_ phase formation began at the temperature of ~1400 °C, while in the chemically non-modified material the α-Al_2_O_3_ phase commenced at the temperature of ~1150 °C.

The experimental results were interpreted using a physical model that quantitatively describes the evolution of 3D structure in three ranges: (I) low temperatures—water desorption and dehydration of amorphous [Al_2_O_3_∙*n*H_2_O], (II) moderate temperatures ≥ 750 °C—surface diffusion activation, (III) elevated temperatures ≥ 1200 °C—sintering of porous structure.

It was established that, up to temperatures of ~1400 °C, no noticeable interaction of the SiO_2_ layer with surface of the 3D substrate took place, while formation of the mullite phase (~2Al_2_O_3_·SiO_2_) began at ~1500 °C. During annealing, the specific surface area of the 3D substrate decreased, which resulted in the increase in SiO_2_ layer thickness from the initial value in raw material of 0.96 ± 0.05 to ~150 ML after annealing at temperatures ≥ 1600 °C.

This work is an extension of the research on chemical and structural transformations of ultraporous monolithic 3D nanomaterials consisting of fibrillar aluminum oxyhydroxides [5,11,12]. The obtained results will be useful in designing new functional 3D nanostructures and nanocomposites.

## Figures and Tables

**Figure 1 nanomaterials-12-03591-f001:**
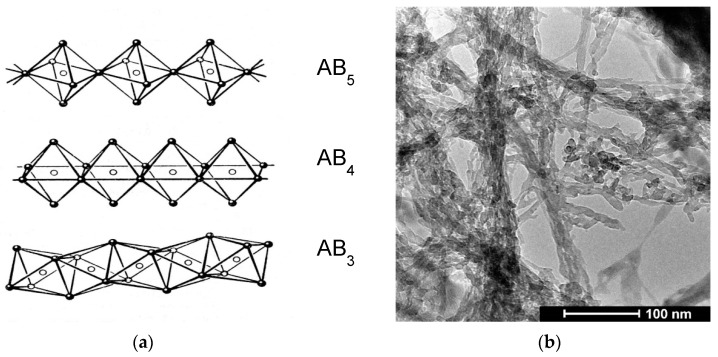
(**a**) Molecular chains of Al^+3^ cations (hollow balls) in anionic octahedral surroundings (solid balls) and corresponding stoichiometry of octahedral chains with the composition ABx. The side contacts and association of molecular chains with fibrils forms a 3D structure of raw PMOA. (**b**) TEM image of raw PMOA. Individual nanofibrils with a diameter of 5–7 nm and bundles of nanofibrils with a diameter of 20–50 nm is clearly seen.

**Figure 2 nanomaterials-12-03591-f002:**
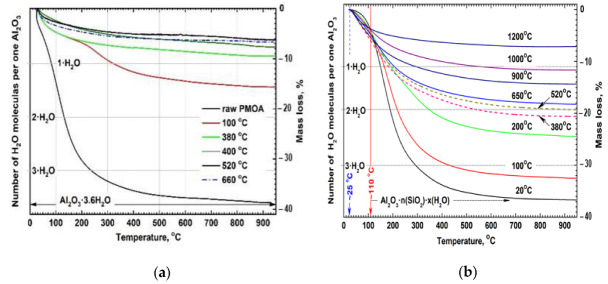
Molecular ratio H_2_O/Al_2_O_3_ and relative mass loss for: (**a**) monolithic raw PMOA sample and the samples pre-annealed at the fixed temperature and cooled down in humid air (except the powder sample PMOA pre-annealed at 660 °C) [11]; (**b**) monolithic PMOA-M sample and the samples pre-annealed at the fixed temperature and cooled down in humid air. From the comparison of the mass loss curves, it follows that a noticeable dehydration of the PMOA-M samples begins at the temperatures ≥110 °C and almost immediately upon heating for the PMOA samples without chemical surface modification.

**Figure 3 nanomaterials-12-03591-f003:**
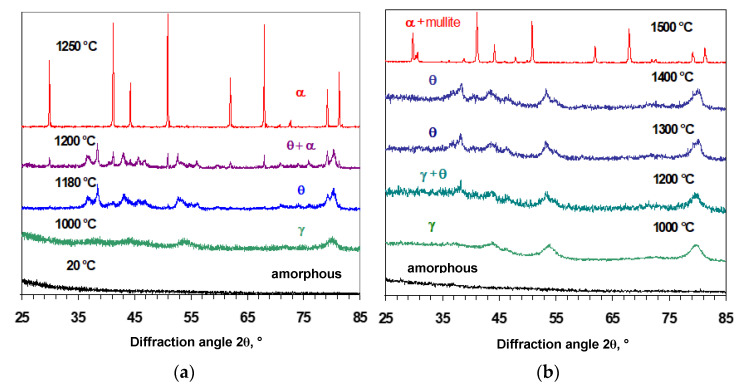
XRD patterns of the samples after isochronous 4 h annealing at fixed temperatures: (**a**) PMOA and (**b**) PMOA-M.

**Figure 4 nanomaterials-12-03591-f004:**
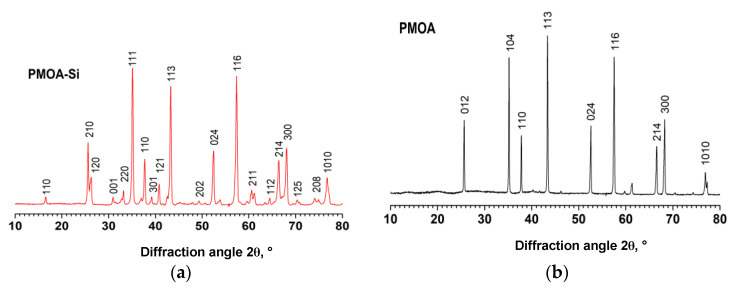
X-ray diffraction patterns of PMOA-M (**a**) and PMOA (**b**) samples after sintering at 1400 °C for 4 h.

**Figure 5 nanomaterials-12-03591-f005:**
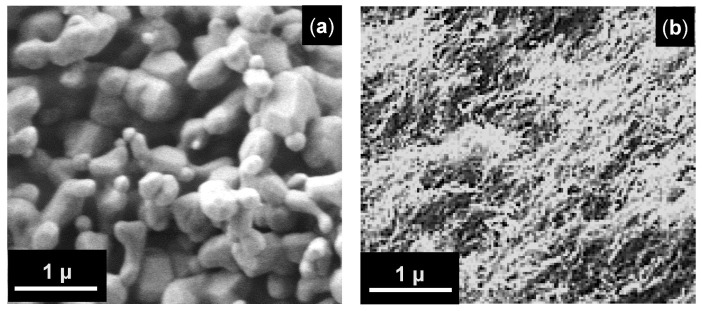
Effect of the annealing in air at the temperature of 1400 °C for 4 h on morphology of the samples: SEM images of (**a**). PMOA, and (**b**). PMOA-M.

**Figure 6 nanomaterials-12-03591-f006:**
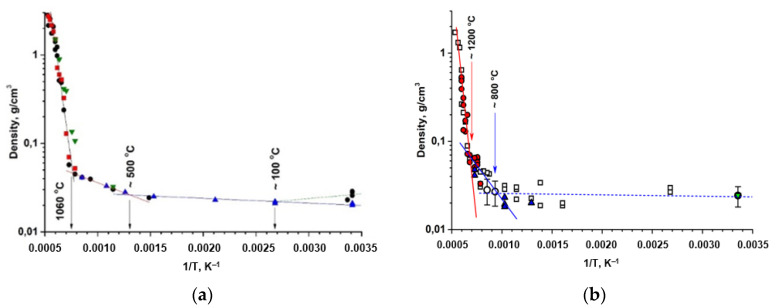
Evolution of the sample volume density as a function of the annealing temperature. The different markers indicated the different series of measurements. The characteristic transition temperatures are marked on the graphs: (**a**) PMOA samples: ~100 ÷ 500 °C—desorption and dehydration; 550 ÷ 1000 °C—the surface diffusion activation range; ≥1060 °C—sintering region [11]. (**b**) PMOA-M samples: ~150 ÷ 750 °C—desorption and dehydration; 750 ÷ 1150 °C—the surface diffusion activation range; ≥1200 °C—sintering region.

**Figure 7 nanomaterials-12-03591-f007:**
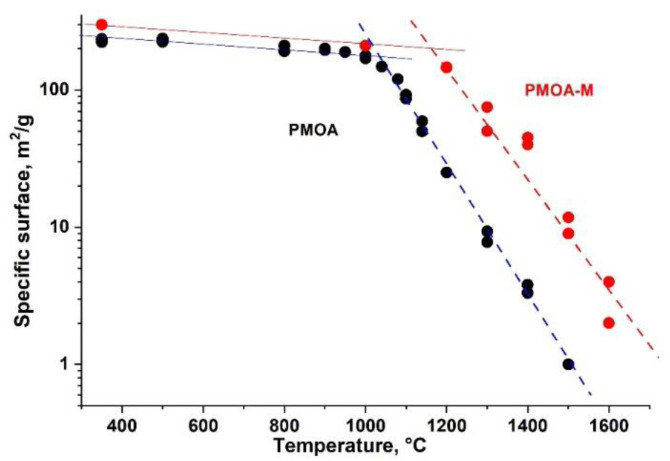
Specific surface area of pure PMOA and PMOA-M at different temperatures in isochronous 4h annealing.

**Figure 8 nanomaterials-12-03591-f008:**
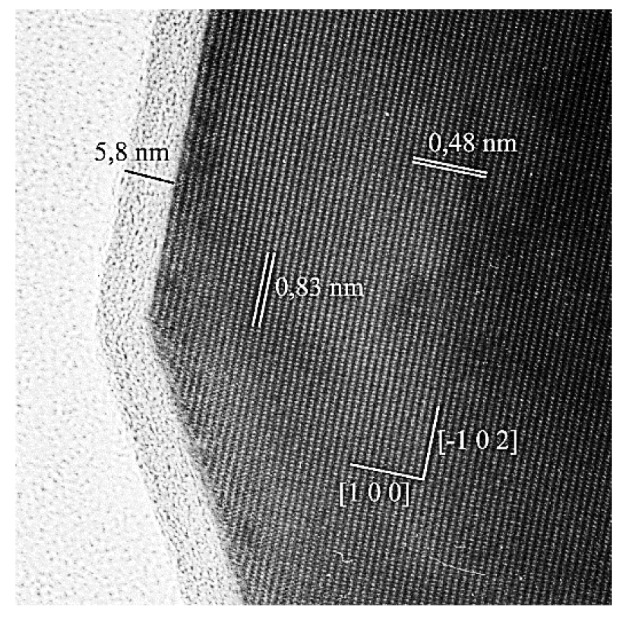
200 kV HRTEM (TOPCON 002B) image of a PMOA-M sample annealed at 1400 °C. A layer of amorphous SiO_2_ of 5.8 nm thickness or 17 ± 2 ML covers the surface of the crystalline Al_2_O_3_ fibril.

**Figure 9 nanomaterials-12-03591-f009:**
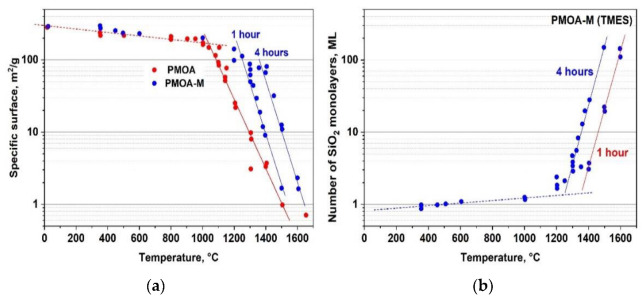
PMOA samples exposed for 1h and 4h to MTM vapor and annealed at the temperatures of 350–1600 °C: (**a**) specific surface variations and (**b**) the thickness of SiO_2_ layer on the fibril surfaces.

**Table 1 nanomaterials-12-03591-t001:** The physical properties of raw PMOA samples [12].

Physical Property	Raw PMOA
Apparent density, g∙cm^−3^	0.018 ± 0.005
Skeletal density, g∙cm^−3^	2.25 ± 0.05
Porosity, %	99 ± 1
Surface area, m^2^∙g^−1^	290 ± 20
Pore volume, cm^3^∙g^−1^	2.95
Average pore diameter, nm	43

**Table 2 nanomaterials-12-03591-t002:** Water content in PMOA-M samples: Al_2_O_3_∙*n*H_2_O∙*x*(SiO_2_), were *n* = *n*_str_ + *n*_ads_.

Molar Ratio H_2_O/Al_2_O_3_	Pre-Annealing Temperature, °C					
	~25 **	100	200	380	520	650
Water total, *n*	~3.6	3.24	2.43	2.06	1.92	1.81
Structural water, *n*_str_	~3.0 *	0.05	1.31	0.55	0.34	0.26
Adsorbed water, *n*_ads_	~0.6*	3.19	1.12	1.51	1.58	1.55

* The formal estimate based on pre-gibbsite stoichiometry: Al_2_O_3_∙3H_2_O; *n*_ads_ value combines excess of the physically adsorbed water and hydrogen bound molecules in PMOA-M. ** Raw PMOA-M.

**Table 3 nanomaterials-12-03591-t003:** Temperatures of phase transitions and the transition temperature shift after isochronous 4 h annealing of PMOA and PMOA-M nanomaterials.

Phase Transition		Transition Temperature, °C		Transition Temperature Shift, °C
		PMOA	PMOA-M	
I-II	Polymolecular polynuclear state → Amorphous phase	<100	≤120	~0
II	Amorphous phase with the elements of pre-boehmite structures (APBS)	≤460	≤850	~400
II-III	Crystallization: APBS → γ-Al_2_O_3_	≤870	~870	~0
III-IV	γ-Al_2_O_3_ → θ-Al_2_O_3_	~1000–1100	~1150–1300	~150
IV-V	θ-Al_2_O_3_ → α-Al_2_O_3_	~1150	≥1400	≥250

**Table 4 nanomaterials-12-03591-t004:** 3D model parameters of PMOA and PMOA-M materials in temperature range 25–1700 °C.

Parameter of the 3D Model	PMOA	PMOA-M
**Low temperature range, °C** The “effective” surface diffusion equation with the initial parameters set: $a_{0}$ = 140 nm, $d_{0}$ = 5 nm, $\rho_{f}$ 2.25 g∙cm^−3^.	≤800 °C	≤1200 °C
Pre-exponent coefficient *D*_0_, m^2^/s	6.0 × 10^−18^	3.5 × 10^−17^
The activation energy of surface diffusion, *E*_a,_ kJ/mol	28	68.5
**High temperature range, °C** Simplified Ivensen equation with the initial parameters set: $a_{0}$ = 140 nm, $d_{0}$ = 5 nm, $\rho_{f}$ = 3.1 g∙cm^−3^.	>800 °C	>1200 °C
Free volume constant (shrinkage rate) *B*_0,_ s^−1^	2.3 × 10^−2^	1.23 × 10^−1^
Activation energy of the sintering *E*_b_, kJ/mol	58	87.5

## Data Availability

The data presented in this study are available on request from the corresponding author.

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
