# Peer review of "Effects of Surface Chemical Modification by Ethoxysilanes on the Evolution of 3D Structure and Composition of Porous Monoliths Consisting of Alumina Hydroxide Nanofibrils in the Temperature Range 25–1700 °C"

_nanomaterials, 2022, doi:10.3390/nano12203591_

Round 1
Reviewer 1 Report
The authors systematically studied the influence of silica monolayer on the surface affects the kinetics of aluminum oxyhydroxide dehydration in the range from 20 to 1700 °C. Overall, this manuscript is well organized, which can be accepted by Nanomaterials after minor revision.
1. Some unreadable codes are observed in Figure 2. Please update it.
2. More discussion regarding the effects of the thickness of SiO2 layers on the surface of PMOA-M should be added. Also, the related characterizations should be provided.
3. How to control the the thickness of SiO2 layer, the authors should discuss it.
4. In figure 8, the 0,48 and 0,83 should be 0.48 and 0.83 nm. Also, the scale bar was missing.
5. English writting should be improved. Also, the authors shoud re-check the manuscript carefully to avoid typos.
Author Response
Reply to Reviewer 1.
- Some unreadable codes are observed in Figure 2. Please update it.
It is not clear to me what "unreadable codes" were observed in Figure 2. Please clarify. In my manuscript, in both versions of the docx and pdf files, all the inscriptions in Figure 2 are readable no worse than the main text.
- More discussion regarding the effects of the thickness of SiO2 layers on the surface of PMOA-M should be added. Also, the related characterizations should be provided.
I have added the following paragraph before describing the quantitative model for the evolution of the PMOA-M structure:
Taking into account the results obtained for PMOA and PMOA-M samples and comparing them in terms of the kinetics of the chemical composition changes (Figure 2), the volume density evolution (Figure 6), structural and phase transformations during annealing (Figure 3), allow us to state that even single monolayers of SiO2 on the surface of PMOA-M lead to: 1) a noticeable inhibition of the dehydration of nanofibrils during annealing; 2) the onset of phase transformations in PMOA-M shifts towards higher temperatures, in particular, the difference exceeds 250 °C for the θ-Al2O3 → α-Al2O3 phase transition during isochronous 4-hour annealing (Table 3); 3) in the temperature range up to 1200 °C, the surface diffusion in PMOA-M samples is much less than for an open surface in PMOA. All of these affect the overall kinetics of structural changes, minimising the morphology changes of nanofibrils and 3D nanostructure remains close to the initial (Figure 5).
Comparison of the main physical parameters characterizing PMOA and PMOA-M materials in the temperature range of 25 – 1700 °C was carried out using the physical model proposed by us and the results are presented in Table. 4.
- How to control the thickness of SiO2 layer, the authors should discuss it.
For quantitative estimates of the average thickness of the SiO2 layer on the surface, we used the total chemical composition of Al2O3∙nH2O∙x(SiO2) data of the PMOA-M nanocomposite, obtained by TGA at temperatures ≤ 1000 °C, and corresponding dependences of the specific surface area or volume density in the temperature range ≤ 1600 °Ð¡; calculations were carried out according to formulas 3.7 - 3.12.
- In figure 8, the 0,48 and 0,83 should be 0.48 and 0.83 nm. Also, the scale bar was missing.
This HRTEM image showing the amorphous SiO2 layer at a crystallite surface was given to me by Leo Mazerolles and Daniel Michel during our collaboration teamwork in CECM CNRS. All markers of the interplanar distances in the Al2O3 crystal and thickness evaluation of the SiO2 layer were performed by Daniel Michel. I would not like to change anything in the picture received from the author. Figure 8 has not been published before and is given by me as evidence that the amorphous SiO2 layer is retained on the Al2O3 surface during annealing in the range from 25 to 1400 °C, regardless of a number of structural-phase transformations of the crystalline phases of nanofibrils: from amorphous Al2O3 nH2O to γ-, θ-, α- and others. A fairly complete description of the similar results published by L. Mazerolles, D. Michel, J.-L. Vignes can be found at [12]
- English writing should be improved. Also, the authors should re-check the manuscript carefully to avoid typos.
I carefully read the manuscript and also used Word's built-in error corrector.
Reviewer 2 Report
This is a very interesting work to investigate the effects of the surface chemical modification by ethoxysilanes on the evolution of PMOA-based materials at high temperature. It shows that the presence of a silica layer on the surface affects the kinetics of aluminum oxyhydroxide dehydration, inhibits diffusion mass transfer and structural phase transformations. This provides an important insight into the development of PMOA-based materials with high performances. The corresponding comments are as follows:
1. The title is too long, please use a concise and creative one, which the readers can get the key of this work easily.
2. What is the abbreviation of PMOA, the authors only give the PMAO in line 47. Is this the same material?
3. In Figure 1, I can not get the relationship of ABx with the TEM image. Please give an explanation of (a) and (b) in the text.
4. The performance difference between PMOA and PMOA-M is mainly about the specific surface area at different temperature, especially at high temperature. The authors might provide more difference in application of the materials, such as mechanistic, heterogeneous catalysis, biodegradation of microorganisms, optics, antenna arrays, etc.
5. Please pay attention to the details of the paper, such as 2 dot in line 31, strange symbol in Figure 3 and 5,8 nm in Figure 8, etc.
In summary, if the authors could revised as the suggestion above, I recommend it to be published.
Author Response
- The title is too long, please use a concise and creative one, which the readers can get the key of this work easily.
I agree that the title of the article is long, however, it accurately and fully reflects the content of the work. It seems to me that any other abbreviation of the title of the work may lead to ambiguity. For example: "Effects of the surface chemical modification by ethoxysilanes on the evolution of 3D structure and composition of porous monoliths consisting of the alumina hydroxides nanofibrils" - it is not clear why surface chemical modification leads to evolution of 3D structure and composition. I tried to shorten the title of the article, but I could not reasonably combine: "surface chemical modification", "evolution of 3D structure and composition", "porous monoliths" and "nanofibrils of alumina hydroxides". I will be very grateful to you for any suggestions to shorten the title of the article and I am ready to discuss them.
- What is the abbreviation of PMOA, the authors only give the PMAO in line 47. Is this the same material?
Thank you for your observation and this remark!
Indeed, I gave the definition of "monolithic blocks of a highly porous 3D nanomaterial based on aluminum oxyhydroxides (PMOA), but I made a mistake and rearranged two letters! I have already corrected this mistake in the manuscript: now PMOA and PMOA-M are everywhere!
- In Figure 1, I can not get the relationship of ABx with the TEM image. Please give an explanation of (a) and (b) in the text.
I added a link [15] regarding the 1D growth of molecular chains ABx and also was changed the caption to Figure 1. (a) Molecular chains of Al+3 cations in anionic octahedral surroundings and corresponding stoichiometry of octahedral chains with the composition ABx, the side contacts and association of molecular chains into fibrils forms a 3D structure of raw PMOA. (b) TEM image of raw PMOA. Individual nanofibrils with a diameter of 5–7 nm and bundles of nanofibrils with a diameter of 20–50 nm is clearly seen.
- The performance difference between PMOA and PMOA-M is mainly about the specific surface area at different temperature, especially at high temperature. The authors might provide more difference in application of the materials, such as mechanistic, heterogeneous catalysis, biodegradation of microorganisms, optics, antenna arrays, etc.
The difference in properties between PMOA and PMOA-M is associated not only with the specific surface area, but also with significant changes in the nanofibril morphology: (1) the average aspect ratio of nanofibril length to its diameter and (2) the structural-phase state of the nanofibril material. This was shown in our previous article: Khodan, A., Nguyen, T. H. N., Esaulkov, M., Kiselev, M. R., Amamra, M., Vignes, J.-L, Kanaev, A. Porous monoliths consisting of aluminum oxyhydroxide nanofibrils: 3D structure, chemical composition, and phase transformations in the temperature range 25 - 1700 °C. J Nanopart Res 2018 20, 194-204. doi:10.1007/s11051-018-4285-4. But I decided not to develop these ideas in the article under review, which would have led to a significant increase in the overall paper volume.
For the same reason, I did not discuss "differences in application of the materials, such as mechanistic, heterogeneous catalysis, biodegradation of microorganisms, optics, antenna arrays, etc.". Although in all the listed applied areas we have some experience and publications. I believe that the main objective of the article is to demonstrate the effects of the influence of a chemically modified surface layer on the evolution and control of the parameters of a porous 3D nanostructure.
- Please pay attention to the details of the paper, such as 2 dot in line 31, strange symbol in Figure 3 and 5,8 nm in Figure 8, etc.
I carefully read the manuscript and corrected: 2 dot in line 31 (it is 1 dot in line 32 now), Figure 3 completely redone and extra numbers next to the y-axes have been removed, and 5,8 nm in Figure 8 is now 5.8 nm. (In the latter case, my error is related to different standards for displaying decimal fractions: in Russia it is a comma (,) and in the West it is a dot (.)… ?).
Reviewer 3 Report
This work discussed the degree of influence of a silica monolayer chemically deposited on the surface of a mon-22 olithic ultraporous nanostructure, consisting of 3D nanofibrils network of aluminum oxyhydroxide, on the evolution of structure and morphology, on the chemical composition and the phase transfor-24 mations under heat treatment. This paper can be considered by Nanomaterials. Advices are shown below.
1. What are the main factors affecting the different morphologies of materials? The growth mechanism needs to be discussed in detail.
2. Some important references are missing, for Prof. Wenjun Zheng’s group in Nankai University.
3. The Figure 3 need to be double checked.
4. In Figure 5, the SEM photograph needs to be enlarged.
5. In Fig. 8, the scale of the projection electron microscope photograph should be added.
6. The advantages of this work will be described by comparing the previous literatures.
Author Response
- What are the main factors affecting the different morphologies of materials? The growth mechanism needs to be discussed in detail.
The original morphology of raw PMOA is the same for both PMOA and PMOA-M materials. The main factor causing a change in the morphology of PMOA and PMOA-M materials is diffusion processes as on their surface as well as in the volume of nanofibrils during annealing of a 3D nanostructure. The article is devoted to the study of the effect of a thin surface SiO2 monolayer on the evolution of a 3D nanostructure and structural phase transformations in the nanofibril material.
The mechanism of 1D growth of nanofibrils followed by the formation of a 3D nanostructure on the surface of Me(Al) liquid-metal phases has not yet been studied. More than 110 years have passed since the first publication : Wislicenus H. Zeitschrift für chemie und industrie der kolloide Kolloid-Z. 2 (1908) XI-XX… The composition and structure of these fibrous aluminas for the first time quite fullyby have been described M. R. Pinnel and J. E. Bennett. Voluminous oxidation of aluminium by continuous dissolution in a wetting mercury film, J. Mater. Sci. 1972 7 1016-1026.
It can be stated with confidence that in our works, for the first time, a quantitative physical model of the evolution of the 3D structure and composition of PMOA and PMOA-M porous monoliths was proposed. As for the mechanism of formation and 1D growth of nanofibrils, we are making attempts to solve this problem together with the theoreticians of the National Research Center Kurchatov Institute. The main postulate of the model we are developing is that the formation of nanofibrils occurs not on the surface of the liquid metal melt, but in the near-surface layers. It is important to note that this 1D growth mechanism is preserved in the temperature range from 20 to 650 °C!
You can get more complete information by viewing my presentation at the link in the cloud: https://cloud.mail.ru/public/XkFA/5NZQUPPvZ
- Some important references are missing, for Prof. Wenjun Zheng’s group in Nankai University.
I found several publications by Prof. Wenjun Zheng from Nankai University related to aluminum oxides:
- Understanding the effect models of ionic liquids in the synthesis of NH4-Dw and gamma-AlOOH nanostructures and their conversion into porous gamma-Al2O3, Chem. Eur. J., 2013, 19(19), 5924-5937.
- Formation of alumina nanocapsules by high-energy-electron irradiation of Na-dawsonite nanorods, Scientific Reports, 2013, 3, (DOI: 10.1038/srep03218)
- One-step ionothermal synthesis of gamma-Al2O3 mesoporous nanoflakes at low temperature, Chem. Commum., 2010, 46(15), 2650-2652.
Unfortunately, I did not find it possible to use these works as references in my article. If you know other publications of Prof. Wenjun Zheng, which are much closer in subject matter to my work, please report to my e-mail : anatole.khodan@gmail.com.
- The Figure 3 need to be double checked.
The Figure 3 was completely redrawn and double checked.
- In Figure 5, the SEM photograph needs to be enlarged.
SEM images in Figure 5 are shown at the same magnification to show the significant difference in nanofibril morphology between PMOA and PMOA-M materials. I believe that changing the scale of images can make it difficult to compare the sizes of nanofibrils.
- In Fig. 8, the scale of the projection electron microscope photograph should be added.
I changed a little Figure 8 and the scale marker was added.
- The advantages of this work will be described by comparing the previous literatures.
This work is a direct continuation of the previously published work : [11] Khodan, A., Nguyen, T. H. N., Esaulkov, M., Kiselev, M. R., Amamra, M., Vignes, J.-L, Kanaev, A. Porous monoliths consisting of aluminum oxyhydroxide nanofibrils: 3D structure, chemical composition, and phase transformations in the temperature range 25 - 1700 °C. J Nanopart Res 2018 20, 194-204. doi:10.1007/s11051-018-4285-4. I wrote about this in 1. Introduction : This work continue our search for the synthesis methods of monolithic porous 3D nanomaterials based on aluminum oxyhydroxides and the studies of the evolution of physical and chemical properties during annealing at temperatures up to 1700 °Ð¡ [11, 12], as well as in part 4. Conclusions: This work is a continuation of the research on the chemical and structural transformations in highly porous monolithic 3D nanomaterials consisting of fibrillar aluminum oxyhydroxides [5, 11, 12]. So, this is not a comparison of "The advantages of this work", but the development of previous scientific research…
Round 2
Reviewer 3 Report
accept
Author Response
The manuscript was substantially edited and corrected taking into account the comments of all reviewers. English language, style and spelling have been corrected and improved. See the latest version of the manuscript 24.09.2022.